# Phosphate starvation decouples cell differentiation from DNA replication control in the dimorphic bacterium *Caulobacter crescentus*

**Joel Hallgren**, **Kira Koonce**, **Michele Felletti**, **Julien Mortier**, **Eloisa Turco**, **Kristina Jonas** *

Science for Life Laboratory and Department of Molecular Biosciences, The Wenner-Gren Institute, Stockholm University, Stockholm, Sweden

* kristina.jonas@su.se

## Abstract

Upon nutrient depletion, bacteria stop proliferating and undergo physiological and morphological changes to ensure their survival. Yet, how these processes are coordinated in response to distinct starvation conditions is poorly understood. Here we compare the cellular responses of *Caulobacter crescentus* to carbon (C), nitrogen (N) and phosphorus (P) starvation conditions. We find that DNA replication initiation and abundance of the replication initiator DnaA are, under all three starvation conditions, regulated by a common mechanism involving the inhibition of DnaA translation. By contrast, cell differentiation from a motile swarmer cell to a sessile stalked cell is regulated differently under the three starvation conditions. During C and N starvation, production of the signaling molecules (p)ppGpp is required to arrest cell development in the motile swarmer stage. By contrast, our data suggest that low (p)ppGpp levels under P starvation allow P-starved swarmer cells to differentiate into sessile stalked cells. Further, we show that limited DnaA availability, and consequently absence of DNA replication initiation, is the main reason that prevents P-starved stalked cells from completing the cell cycle. Together, our findings demonstrate that *C. crescentus* decouples cell differentiation from DNA replication initiation under certain starvation conditions, two otherwise intimately coupled processes. We hypothesize that arresting the developmental program either as motile swarmer cells or as sessile stalked cells improves the chances of survival of *C. crescentus* during the different starvation conditions.

## Author summary

Bacteria frequently encounter periods of nutrient limitation. To ensure their survival, they dynamically modulate their own proliferation and cellular behaviors in response to nutrient availability. In many *Alphaproteobacteria*, progression through the cell cycle is tightly coupled to morphological transitions generating distinct cell types. Here, we show how

**Data Availability Statement:** Detailed descriptions of the image analysis workflow and training datasets, as well as underlying numerical data for all graphs and summary statistics have been made

publicly available at the Swedish National Data Service (SND-ID: 2023-202), accessible at: https://doi.org/10.58141/wfc6-qh32. All other relevant data are within the manuscript and its Supporting Information files.

**Funding:** The study was funded by grants from the Swedish Research Council (2016-03300 to KJ, 2020-03545 to KJ), a future leaders grant from the Swedish Foundation for Strategic Research (SSF, FFL15-0005 to KJ), the European Union's Horizon 2020 research and innovation programme under the Marie Skłodowska-Curie grant agreement no. 797801 to MF as well as funding from the Strategic Research Areas (SFO) program distributed through Stockholm University to KJ. The funders had no role in study design, data collection and analysis, decision to publish, or preparation of the manuscript.

**Competing interests:** The authors have declared that no competing interests exist.

starvation for either of the major nutrients carbon, nitrogen, or phosphorus affects this coupling between key cell cycle events and cell differentiation in the model bacterium *Caulobacter crescentus*. All three starvation conditions prevent cell proliferation by blocking DNA replication initiation. However, while carbon and nitrogen exhaustion cause cells to arrest the cell cycle as non-replicating motile cells, phosphorus starvation leads to accumulation of non-replicating sessile stalked cells. Our data indicate that starvation-dependent differences in (p)ppGpp signaling account for these different starvation responses. Together, our work provides insights into the mechanisms that allow bacteria to modulate their developmental program in response to changing environmental conditions.

## Introduction

At the onset of starvation, bacteria rapidly shut down reproductive functions and modify their behavior to ensure survival. Behaviors such as cell motility and chemotaxis allow cells to actively track down nutrients, while surface colonization and biofilm formation help cells to persist by optimizing the utilization of preexisting resources [1,2]. Simultaneously, energetically costly and vulnerable proliferative functions, like DNA replication and cell division, need to be shut down during starvation. Yet, precisely how bacteria coordinate the regulation of cellular proliferation, development, and behavior to overcome starvation remains incompletely understood.

The alphaproteobacterium *Caulobacter crescentus* offers an attractive model to study bacterial adaptation to nutrient limitation. *C. crescentus* and closely related lineages inhabit freshwater environments [3] that are often limited for key nutrients, like phosphorus and nitrogen [4–8], and show strong fluctuations in nutrient availability as a function of seasonality [9,10]. Additionally, the life cycle of *C. crescentus* comprises two morphologically and behaviorally distinct cell types, providing a powerful system for studying bacterial proliferation and cell development (**Fig 1A**) [11,12]. The reproducing form of *C. crescentus*—the sessile stalked cell type—is named by its thin unipolar cell envelope extension, from the tip of which a holdfast is anchored that permanently attaches the cell to a surface [3]. Although the function of the stalk is controversial, it is generally considered to improve nutrient uptake [13,14]. The stalked cell divides asymmetrically to generate motile swarmer cell offspring at the cell pole opposite of the stalk. Swarmer cells are chemotactic and able to locate nutrients and favorable conditions [15]. However, they are arrested in the $G_1$ phase and do not enter S phase until their eventual differentiation into stalked cells [16].

In *C. crescentus*, DNA replication initiation is controlled by the opposing activities of two essential cell cycle regulators: the highly conserved DNA replication initiator protein DnaA and the response regulator CtrA [17]. Like other bacteria, *C. crescentus* requires DnaA in the ATP-bound state to unwind the two DNA strands at the chromosomal replication origin and to recruit the DNA replication machinery. ATP hydrolysis during DNA replication initiation yields the ADP-bound inactive form of DnaA and prevents re-initiation of DNA replication before cell division [18–20]. While the regulation of DnaA activity limits DNA replication initiation to once per cell cycle, CtrA is required to block DNA replication initiation specifically in swarmer cells [17]. CtrA binds the origin and thereby prevents DnaA-mediated replication initiation [21–23]. During the swarmer-to-stalked (SW→ST) cell differentiation, CtrA is inactivated through dephosphorylation and degradation [24,25], which grants DnaA access to the origin, allowing DNA replication to initiate [23]. After DNA replication has initiated, CtrA is

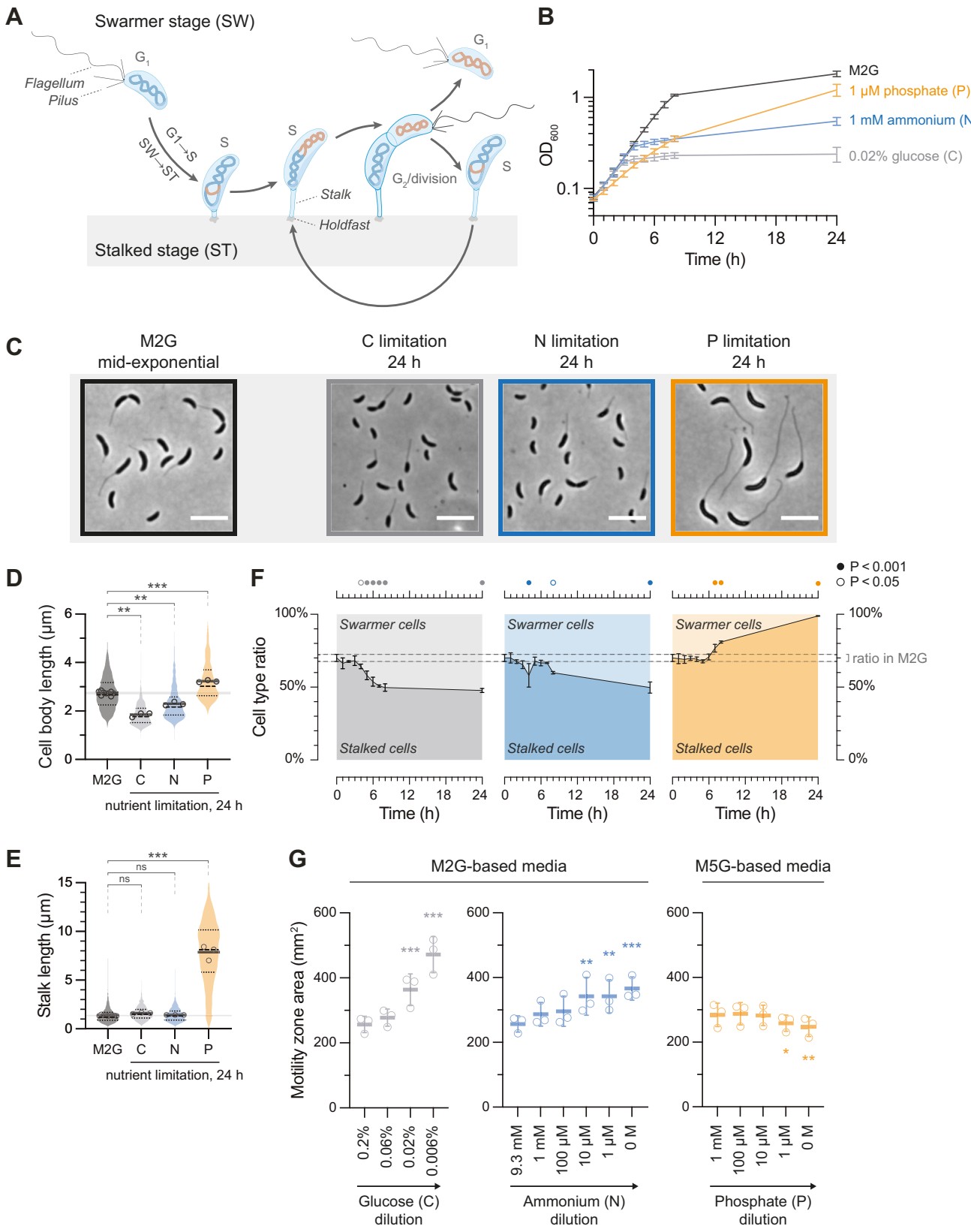

**Fig 1. *Caulobacter crescentus* arrests development as stalked cells under P starvation, but as swarmer cells under C and N starvation. (A)** Schematic representation of the dimorphic life cycle of *C. crescentus*, depicting cell morphological features and chromosome replication. **(B)** Growth curves of wild-type *C. crescentus* after shifting exponentially growing cells from nutrient-replete M2G minimal medium back into the same medium (black; n = 7), or into C-limited (n = 11), N-limited (n = 12), or P-limited (n = 14) minimal media. Data points show mean ± standard deviation (SD). **(C)** Phase-contrast micrographs of cells sampled from an M2G culture in mid-exponential phase, and cells sampled after 24 hours in nutrient-limited media. Scale bars: 5 μm. **(D)** Violin plot of lengths of cells from M2G cultures in mid-exponential phase (450 randomly selected cells for each of the nine biological replicates) or from nutrient-limited cultures after 24 hours (450 randomly selected cells in three biological replicates, per condition). Circles show the mean cell length for each replicate and thick horizontal lines show the average of these means. The extended horizontal line emphasizes the average nutrient-replete cell length. Asterisks depict significant differences determined by an ANOVA test followed by Bonferroni correction of P values from pairwise post hoc comparisons, comparing starved populations to nutrient-replete (M2G); 'ns': non-significant; *: P < 0.05; **: P < 0.01; ***: P < 0.001. **(E)** Violin plot of stalk lengths as shown for (D), but with each biological replicate encompassing 300 cells. **(F)** The proportions of stalked cells to swarmer cells in M2G mid-exponential phase (t = 0 h; nine biological replicates) and after shifting cells to nutrient-limited media (three biological replicates per condition). For each biological replicate, 537–4745 cells were counted (average: 2023 cells). Data points show mean ± SD. Dashed lines emphasize the mean cell type ratio ± SD during mid-exponential phase in M2G medium (t = 0 h). Circles in the top panel depict significant differences determined by an ANOVA test, followed by Bonferroni correction of P values from pairwise post hoc comparisons, comparing cell type ratios at each time point to the initial ratio (t = 0 h). **(G)** Motility of cells in minimal medium soft agar containing the indicated concentrations of glucose, ammonium, and phosphate. Means ± SD of the motility zone area from three independent experiments are shown. Asterisks depict significant differences determined by an ANOVA test, followed by Bonferroni correction of P values from pairwise post hoc comparisons, comparing each nutrient concentration to the nutrient-replete concentration; *: P < 0.05; **: P < 0.01; ***: P < 0.001.

resynthesized and phosphorylated in the predivisional stalked cell [25–28], where it serves as an essential regulator of swarmer cell biogenesis and cell division [29,30]. The interplay between DnaA and CtrA at the origin of replication ensures tight coupling between the SW→ST cell differentiation and DNA replication initiation under optimal conditions.

Previous studies have already provided some insights into how cell cycle progression and development in *C. crescentus* are regulated in response to nutrient limitation. DnaA was found to be strongly downregulated during different starvation conditions [31–35]. Under C starvation, it was recently shown that DnaA clearance is mediated by inhibition of DnaA translation in combination with Lon-mediated proteolysis [34]. The signaling molecules guanosine penta- and tetraphosphate (collectively referred to as [p]ppGpp) have also previously been implicated in the starvation-induced clearance of DnaA [32]. However, later work showed that the downregulation of DnaA under C starvation does not require (p)ppGpp [33]. In *C. crescentus*, (p)ppGpp are produced and degraded by the bifunctional Rel enzyme (also known as SpoT), which is regulated by intracellular glutamine levels [36], triggering (p)ppGpp accumulation during C or N limitation [37]. Although (p)ppGpp do not seem to play a critical role in the starvation-induced downregulation of DnaA abundance, they were shown to inhibit the SW→ST cell differentiation and affect the signaling pathways controlling CtrA [38–42].

While several studies have investigated the regulation of DnaA and CtrA under C starvation [31–34,40,43], and to some degree under N starvation [31,43], less is known about how cell differentiation and DNA replication initiation are coordinated under P starvation. However, this condition is well-known to lead to the accumulation of stalked cells with dramatically elongated stalks [44,45] and has served as a model condition to study features of the *C. crescentus* stalk [35,44,46–48]. This accumulation of stalked cells is characteristic to P starvation, and contrasts the extension of the swarmer stage under N and C starvation [32,38,40,43]. Thus, there must be differences in how information on C, N, and P availability affects the coordination between cell cycle progression and development in *C. crescentus*.

To better understand these differences, we analyzed cell cycle progression and development during starvation for these distinct macronutrients by combining a machine learning-based imaging approach with molecular and cell biological assays. Our results show that although DnaA and DNA replication initiation are regulated by a common mechanism, the SW→ST cell differentiation is controlled differently under these conditions. This results either in the accumulation of motile swarmer cells under C and N starvation, or in pure populations of non-replicating sessile stalked cells under P starvation. Our data suggest that these distinct outcomes can be explained by

a (p)ppGpp-dependent mechanism that halts cell differentiation specifically under C and N starvation, but not under P starvation. Finally, we demonstrate that inability of P-starved stalked cells to complete the cell cycle is caused by absence of DnaA and DNA replication.

## Results

### Distinct starvation conditions arrest *C. crescentus* development at different morphological stages

To better understand how the developmental program of *C. crescentus* is modulated in response to distinct forms of nutrient depletion, we quantified changes in growth, cell size, and morphology in response to C, N, or P limitation. Nutrient-specific starvation was induced by shifting exponentially growing cells from minimal glucose medium (M2G) to minimal media with reduced concentrations of either glucose, ammonium, or phosphate (C, N, or P sources respectively) (**Fig 1B**). Cultures transferred to the C or N starvation media continued to grow for 3–4 hours with the same doubling time as in M2G before abruptly entering a starvation-induced growth arrest upon nutrient exhaustion (**Fig 1B**). The C or N starvation-induced growth arrest was accompanied by a significant reduction in cell size (**Fig 1C and 1D**). In contrast, P-starved cultures slowed down growth immediately after the media shift, but continued to grow throughout the experiment reaching a similar maximum optical density as a control M2G culture after about 48 hours (**Figs 1B and S1A**). Consistent with previous reports [45], P-starved cells increased in cell length (**Figs 1C, 1D,** and **S1B**) and dramatically elongated their stalks (**Fig 1E**). Microscopic analysis also clearly showed that the vast majority of P-starved cells became stalked cells, while C and N starved cultures consisted of a mix of cell types (**Fig 1C**).

To thoroughly quantify relative changes in the proportion of swarmer cells to stalked cells during the course of C, N, and P starvation, we developed a machine learning-based image analysis approach employing the image analysis tools ilastik [49] and MicrobeJ [50] that enabled the automatic annotation of cell types (see Materials and Methods). Using this method on phase-contrast microscopy images revealed that in nutrient-replete exponentially growing cultures, stalked cells comprised approximately 70% of the total population (**Fig 1F**; t = 0 h) and that this fraction of stalked cells significantly decreased to approximately 50% in the C- or N-starved cultures within 24 h, yielding a net swarmer (stalkless) cell accumulation. This result is consistent with the notion that these starvation conditions delay cell differentiation from swarmer cells to stalked cells [32,38,40,43]. In sharp contrast, virtually all cells experiencing P limitation exhibited stalks after 24 hours (**Fig 1F**), indicating that these cells can still undergo cell differentiation into stalked cells, but are then unable to complete the cell cycle with cell division to generate new swarmer cells. The starvation-dependent accumulation of different cell types correlated with corresponding changes in soft-agar motility. While limitation for glucose or ammonium in the soft agar significantly enhanced motility, which is consistent with an accumulation of motile swarmer cells, a slight decrease in motility was observed in soft agar with limiting phosphate concentrations (**Figs 1G** and **S2**). Altogether, these results show that different forms of starvation arrest cellular development at distinct stages. While C and N starvation restrict the SW→ST cell differentiation, leading to an accumulation of motile swarmer cells, P-starved cells are still able to undergo the SW→ST cell differentiation, but then cannot divide to produce new swarmer cells.

### Phosphate starvation decouples the $G_1 \rightarrow S$ transition from the SW→ST cell differentiation

Because the SW→ST cell differentiation in the *C. crescentus* life cycle is tightly coupled to a change in the DNA replication status (**Fig 1A**), we analyzed the effects of C, N, and P

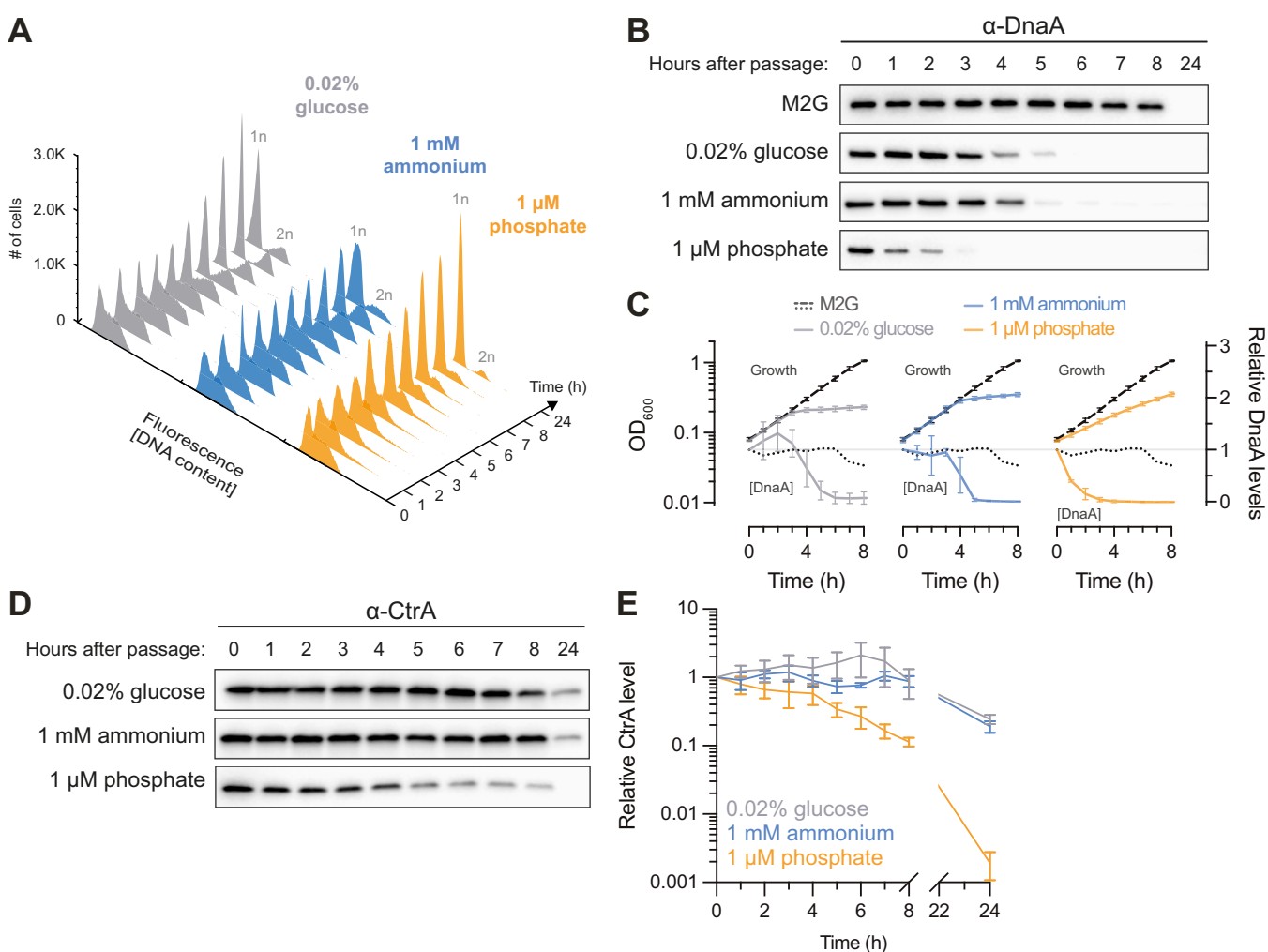

**Fig 2. DNA replication initiation is arrested under C, N, and P starvation. (A)** DNA content histograms of cells in mid-exponential phase in M2G before being shifted to nutrient-limited media (t = 0 h) and of cells incubated in C-limited (gray), N-limited (blue), or P-limited (orange) media over time, as measured by flow cytometry. Each separate histogram represents 30 000 cells. 1n and 2n denote one and two chromosome equivalents respectively. **(B)** Immunoblots of DnaA from nutrient-replete cultures in steady state (t = 0 h) and after shifting cells to nutrient-limited media, or after a mock-shift into nutrient-replete minimal medium (M2G). **(C)** Quantification of DnaA immunoblots done as in (B), alongside growth curves reproduced from **Fig 1A**. Means of relative band intensities (normalized to t = 0 h) from three independent replicates are shown with standard deviations; in case of the mock-shifted M2G culture, data from one replicate are shown. **(D)** Immunoblot of CtrA from nutrient-replete cultures in steady state (t = 0 h) and after shifting cells to nutrient-limited media. **(E)** Quantification of CtrA immunoblots done as in (D), from either four (C-limited) or three (N-/P-limited) independent replicates, with standard deviations.

starvation on the replication status of cells and followed changes in DNA content over the course of starvation by flow cytometry. Despite the different effects of the three starvation conditions on cell differentiation, all three conditions resulted in a block of DNA replication initiation, reflected in an accumulation of cells harboring one (1n) or two (2n) fully replicated chromosomes, with the large majority of cells harboring a single chromosome (**Fig 2A**), consistent with previous reports [33,35,36].

DNA replication initiation in *C. crescentus* is directly regulated by the opposing activities of the replication initiator DnaA and the response regulator CtrA [17]. To compare their regulation under the three starvation conditions, we monitored their abundance by immunoblot analysis. These experiments revealed a strong and rapid clearance of DnaA under all three starvation conditions, coinciding with the starvation-induced decline in growth (i.e., 3–4 hours

after media shift in the case of C and N limitation and within the first hour after media shift in the case of P limitation) (**Fig 2B and 2C**). In contrast to DnaA, CtrA was largely maintained under the different starvation conditions, especially during C and N starvation (**Fig 2D and 2E**). However, during P starvation, CtrA remained at high levels during the first four hours after media shift, whereafter CtrA levels began to drop slowly until its clearance between 8 and 24 hours (**Fig 2D and 2E**). The downregulation of CtrA under P starvation coincided with the accumulation of stalked cells (**Fig 1F**) and is likely a consequence of blocked DNA replication and segregation, which was recently shown to prevent activation of the CtrA signaling pathway [28].

Together, our data demonstrate that *C. crescentus* arrests DNA replication initiation under all three starvation conditions despite these conditions resulting in strikingly different cell type proportions. This indicates that *C. crescentus* decouples SW→ST cell differentiation from the G$_1$→S transition during specific stress conditions, such as P starvation, two processes which are otherwise intimately linked. Furthermore, our results indicate that clearance of DnaA is a key regulatory step in the response to all three starvation conditions.

## DnaA is cleared via translation inhibition under diverse forms of starvation

Previous work showed that DnaA is cleared at the onset of C starvation through a combination of DnaA synthesis inhibition and constitutive Lon-mediated proteolysis [33]. Furthermore, in a recent study we reported that this C starvation-induced inhibition of DnaA synthesis is mediated by the amino acid sequence of the N-terminal region of DnaA (Nt$_{DnaA}$), which reduces *dnaA* translation elongation in response to C unavailability [34]. We therefore wondered if DnaA clearance during N and P starvation depends on a similar mechanism. First, we assessed the contribution of DnaA proteolysis to the starvation-mediated clearance of DnaA by monitoring the *in vivo* stability of DnaA at the onset of C, N, and P starvation. Although DnaA was unstable, the rate of degradation was not markedly faster under any of the starvation conditions than under nutrient-replete conditions (**Fig 3A and 3B**). Consistent with DnaA being degraded by the Lon protease under all three starvation conditions, as shown previously for C starvation [33], DnaA was stabilized in a *lon* knockout mutant also under N and P starvation (**S3A and S3B Fig**). Consequently, DnaA clearance was impaired under C, N, and P starvation (**S3C–S3E Fig**). Notably, in comparison to C and N-starved cells, P-starved cells were still able to efficiently clear cellular DnaA (**S3D Fig**), and elicit a G1 arrest (**S3F and S3G Fig**). This can be explained by DnaA being diluted as a result of continued growth, even in absence of efficient DnaA proteolysis. Taken together, these results indicate that although DnaA is actively degraded by the Lon protease, a change in proteolysis rate cannot account for the rapid DnaA clearance under these conditions.

Next, to see if DnaA synthesis is regulated at the level of protein synthesis and to investigate the contribution of Nt$_{DnaA}$ to this regulation, we made use of a previously established plasmid-borne DnaA translation reporter system [34]. In this system, the upstream regulatory region of *dnaA*, encompassing the promoter, the 5' untranslated region (5'UTR$_{dnaA}$), and the first 26 codons encoding the DnaA N-terminus (Nt$_{DnaA}$) are translationally fused to the enhanced green fluorescent protein (eGFP) (**Fig 3C**). Monitoring of eGFP levels as a proxy for DnaA synthesis, in cells subjected to C, N, or P limitation, showed that under all three starvation conditions eGFP accumulation was repressed as cultures exited steady-state growth upon nutrient exhaustion (**Fig 3D**; Nt). Importantly, when using a construct specifically lacking the Nt$_{DnaA}$ sequence while carrying the same *dnaA* promoter and 5'UTR$_{dnaA}$, eGFP continued to accumulate past the onset of starvation to a much greater level (**Fig 3D**; ΔNt). These results indicate

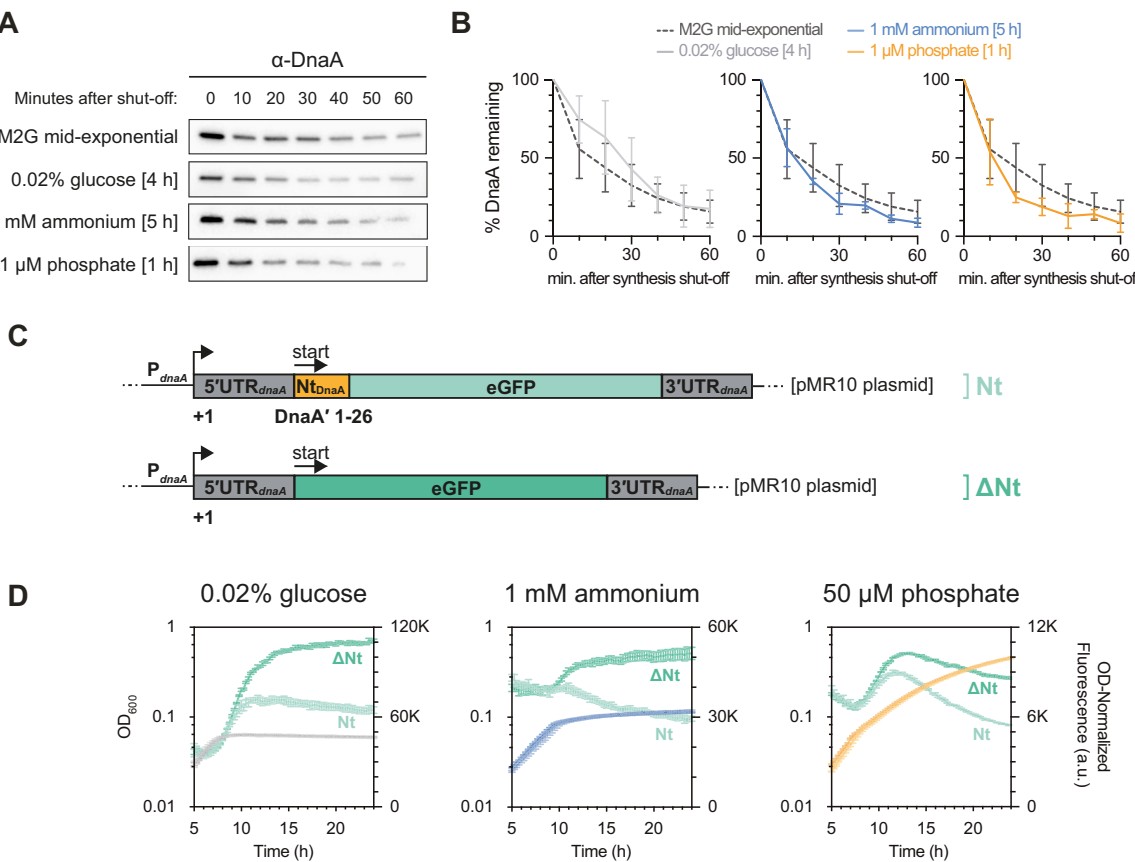

**Fig 3. The replication initiator DnaA is cleared via translation inhibition under diverse forms of starvation. (A)** DnaA *in vivo* stability assay in cells in mid-exponential phase in M2G and in cells experiencing C, N, or P starvation. **(B)** Quantification of DnaA *in vivo* stability assays done as shown in (A), from either ten (M2G mid-exponential phase), four (C-limited) or three (N-/P-limited) independent replicates, with error bars showing standard deviations. **(C)** Schematic depiction of *dnaA* expression reporter constructs from Felletti et al. [34]. **(D)** Growth curves (gray, blue, and orange) and $OD_{600}$-normalized eGFP fluorescence (green) of cells harboring the *dnaA* expression reporter constructs depicted in (C), after shift to C-, N-, or P-limited media. Light hues: the $Nt_{dnaA}$-eGFP construct; dark hues: construct with eGFP alone. Note that the growth curves of the two strains largely overlap.

that the mRNA sequence encoding the N-terminus of DnaA is indeed required for the starvation-induced inhibition of DnaA translation in cells subjected to N or P starvation. The continued increase in optical density after the onset of P starvation is largely due to extensive stalk elongation [35], which likely explains why the optical density eventually outpaces eGFP synthesis under this particular condition. Moreover, as shown previously for C starvation [34], this translation inhibition was independent of the upstream 5′UTR, as replacing the native 5′UTR$_{dnaA}$ with the *lac* operon UTR (5′UTR$_{lac}$) or an artificial UTR (5′UTR$_{6/13}$) retained the effect (**S4 Fig**). These results strongly suggest that DnaA synthesis is inhibited during all three forms of starvation via a shared mechanism involving the N-terminus of DnaA.

## (p)ppGpp block the morphological differentiation into stalked cells under C and N starvation, but not under P starvation

If DNA replication initiation is regulated by a shared mechanism under C, N and P starvation, how do cells starved for these different macronutrients arrest in distinct morphological stages? Since previous work has associated (p)ppGpp accumulation with an extension of the swarmer state [38,39], we analyzed a Δ*rel* mutant, lacking the sole enzyme producing and hydrolyzing

(p)ppGpp in *C. crescentus*. Consistent with a role of (p)ppGpp in inhibiting SW→ST cell differentiation, the C and N starvation-induced changes in cell type ratios were greatly weakened or completely abolished, respectively, in this strain (**Fig 4A**). Instead of accumulating swarmer cells during C and N starvation, as seen for the wild type, the Δ*rel* strain maintained nearly stable cell type ratios throughout the C and N starvation experiments. Correspondingly, the increase in motility observed for C- and N-starved wild-type cells was largely lost in the Δ*rel* mutant (**Figs 4B** and **S2**), and the starvation-dependent reduction in cell size was completely abolished in the Δ*rel* mutant for N starvation and decreased for C starvation (**Figs 4C** and **S5A**). In addition to affecting the developmental responses to C and N starvation conditions, the Δ*rel* mutation also mildly affected the replication status of cells under C and N starvation (**Fig 4D**). C-starved Δ*rel* cells showed an increased proportion of cells with two chromosomes, indicating a delay in cell division in addition to the inhibition of DNA replication initiation, as previously observed [33]. Under N-starvation conditions, the Δ*rel* mutant showed an increased number of cells exhibiting a chromosome content between 1n and 2n, indicating that these cells harbored incompletely replicated chromosomes. Notably, despite these Rel-dependent effects on the replication status of starved cells, rapid DnaA clearance was still observed in nitrogen-starved Δ*rel* cells similar to the wild type (**S5B and S5C Fig**), as has been shown for carbon starvation [33]. This underscores the notion that DnaA clearance under these starvation conditions does not depend on (p)ppGpp.

Strikingly, in contrast to the observed Rel-dependent phenotypes under C and N starvation, the response to P starvation was essentially unaffected by the Δ*rel* mutation. Under this condition, Δ*rel* cells fully differentiated into stalked cells (**Fig 4A**), albeit with slightly different kinetics, and exhibited motility responses to P availability similar to wild-type cells (**Fig 4B**). Furthermore, P starvation of Δ*rel* cells yielded populations with DNA content indistinguishable from wild-type (**Fig 4D**) and showed a similar P starvation-induced increase in cell length as the wild type (**Figs 4C** and **S5A**). These results demonstrate that in contrast to the responses to C and N starvation, the P starvation-induced DNA replication and developmental arrest is independent of (p)ppGpp. This finding is consistent with previous data that (p)ppGpp levels increase during C and N starvation, but not under P starvation [37], and suggests that presence or absence of (p)ppGpp-mediated regulation accounts for the different morphological responses observed under distinct starvation conditions. In other words, while (p)ppGpp accumulation under N and C starvation prevents swarmer cells from differentiating into stalked cells, low (p)ppGpp levels under P starvation allow unhindered differentiation of swarmer cells into stalked cells. Since DNA replication initiation is blocked and DnaA cleared under P starvation in the same way as under N and C starvation, P-starved cells arrest the cell cycle as stalked cells with single chromosomes.

## P-starved stalked cells are unable to complete the cell cycle due to limiting availability of DnaA

Based on our data indicating that the response to P starvation is independent of (p)ppGpp (**Fig 4**), we wondered if the block of DNA replication initiation via DnaA downregulation is the main cause of cell cycle arrest under this form of starvation. Consistent with this idea, P-starved cells resemble the phenotype of genetically engineered DnaA depletion under optimal conditions; both conditions lead to an accumulation of stalked cells that are $G_1$-arrested and filamentous (**Fig 5A**) [51]. To directly address the possibility that DnaA becomes limiting for cell cycle progression under P starvation, we tested if overexpressing *dnaA* can override the P starvation-induced arrest as $G_1$-arrested stalked cells. For this we made use of a strain having its sole native copy of *dnaA* being controlled by the IPTG-inducible *lac* promoter from *E. coli*

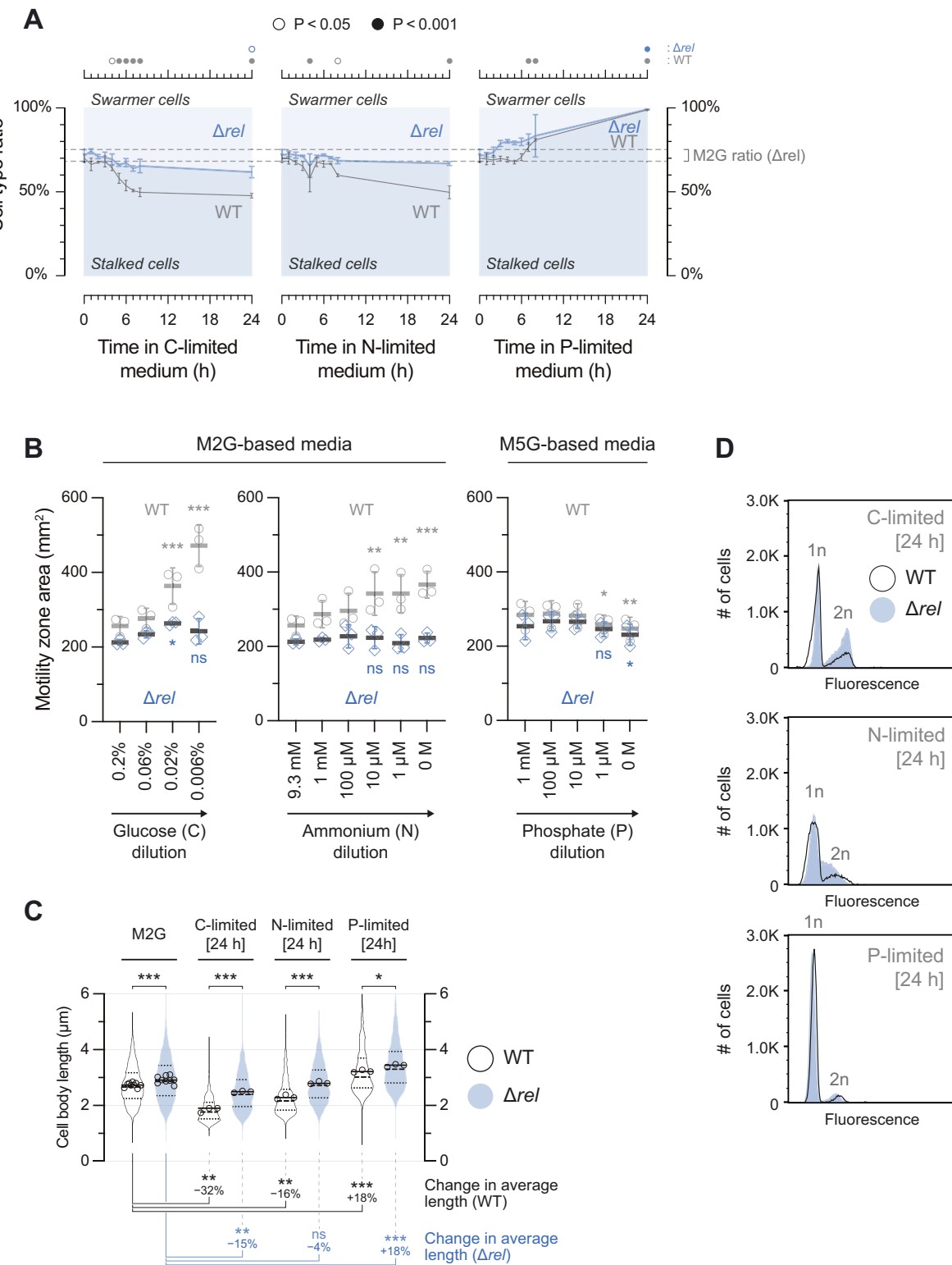

**Fig 4. (p)ppGpp do not orchestrate the P starvation response. (A)** The proportions of stalked cells to swarmer cells in Δ*rel* (Δ*spoT*) cultures subjected to nutrient limitation (blue) as done for the wild type (WT) in **Fig 1F**. Data for wild-type cultures (gray) are included for comparison. For Δ*rel*, 460–3069 cells were counted per biological replicate (average: 1425 cells). Dashed lines emphasize the mean cell type ratio ± SD of Δ*rel* cultures during mid-exponential phase in M2G medium (t = 0 h). Circles in the top panel depict significant differences determined by an ANOVA test, followed by Bonferroni correction of P values from pairwise post hoc comparisons, of cell type ratios at each

time point to the initial ratio (t = 0 h), for each strain separately. **(B)** Motility of Δ*rel* cells as shown for the wild type in **Fig 1G**. Wild-type measurements are overlaid for comparison. "ns" highlights nutrient concentrations where Δ*rel* cells did not exhibit a significant motility difference in comparison to nutrient-replete conditions, but where wild-type cells did. **(C)** Violin plot of cell length measurements, essentially as shown in **Fig 1D** comparing the Δ*rel* mutant to wild-type. Asterisks at the top depict significant differences comparing Δ*rel* to the wild type at each condition. The bottom panel shows the percent change in average cell length, and depicts significant differences, comparing each starvation condition to nutrient-replete conditions (M2G), for each strain separately. Statistical significance was determined by ANOVA tests followed by Bonferroni correction of P values from pairwise post hoc comparisons; 'ns': non-significant; *: $P < 0.05$; **: $P < 0.01$; ***: $P < 0.001$. **(D)** Histograms of the DNA content of 30 000 cells as measured by flow cytometry, comparing the Δ*rel* mutant (blue) to wild-type (black lines) after 24 h in nutrient-limited media. The outlines of the wild-type histograms shown in **Fig 2A** are overlaid for comparison. 1n and 2n denote one and two chromosome equivalents respectively.

[33]. Incubation of this strain in minimal medium in the presence of 50 μM IPTG yielded DnaA levels and cell morphologies close to those of wild-type cells under the same conditions, whereas *dnaA* overexpression could be achieved with 1 mM IPTG (**S6A–S6C Fig**).

To induce *dnaA* overexpression over the course of starvation, we shifted exponentially growing cells from medium containing 50 μM IPTG into C, N, or P-limited media containing 1 mM IPTG. Immunolotting confirmed that under all three conditions IPTG-induced *dnaA* overexpression boosted DnaA synthesis enough for DnaA to be detectable long past the onset of starvation (**Fig 5B**). Under C or N starvation, cell type ratios and chromosome content were not notably affected by the overexpression of *dnaA* (**Figs 5C–5E** and **S7A**), supporting the idea that multiple mechanisms contribute to the cell cycle arrest under these starvation conditions. Strikingly, however, overexpression of *dnaA* from $P_{lac}$ strongly affected the cell type ratio and DNA replication status of P-starved cells. While wild-type cells exhibited only 0.9% swarmer cells and 1.4% predivisional stalked cells after 24 hours in IPTG-supplemented P-limited medium (compare to the 1.0% and 1.4% respectively in absence of IPTG), these fractions were 5.9% and 33.5% in the $P_{lac}$-*dnaA* strain (**Fig 5C** and **5F**). Moreover, flow cytometry revealed a high proportion of cells with a chromosome content of $> 1n$, all throughout the starvation response (**Fig 5G**). The restored ability of *dnaA*-overexpressing P-starved cells to replicate and constrict strongly suggests that DnaA levels are normally limiting under P starvation, arresting cells in the $G_1$ phase. Given that SW→ST cell differentiation is unhindered under P starvation, nascent swarmer cells arising from the *dnaA* overexpression-induced cell division are still likely to promptly differentiate into stalked cells, which could explain why only 5.9% of P-starved $P_{lac}$-*dnaA* cells were stalkless at 24 hours (**Fig 5F**). Interestingly, *dnaA*-overexpressing cells also exhibited significantly shorter stalks (**Figs 5C** and **S8**). This result might be explained by the repositioning of peptidoglycan synthesis and remodeling enzymes from the stalk to the division septum. Partial restoration of DNA replication and cell division was even seen when *dnaA* expression was induced by IPTG addition after 24 hours of P starvation (**S7B and S7C Fig**). However, IPTG-induced *dnaA* expression resulted only in slow and weak DnaA accumulation, likely because cells were severely starved at this point (**S7D Fig**).

Based on these results we conclude that low DnaA availability and the resulting block of DNA replication initiation is the main reason for the developmental arrest of *C. crescentus* as $G_1$ stalked cells. The tight control of DnaA in conjunction with continued DNA replication elongation, cell division—and importantly SW→ST cell differentiation—converts the whole cell population into $G_1$-arrested stalked cells upon P starvation.

## Discussion

### Decoupling the SW→ST cell differentiation from the $G_1$→S transition

When facing starvation conditions, bacteria must downregulate proliferative functions and alter their lifestyle to ensure their survival. Here we show that the freshwater bacterium *C.*

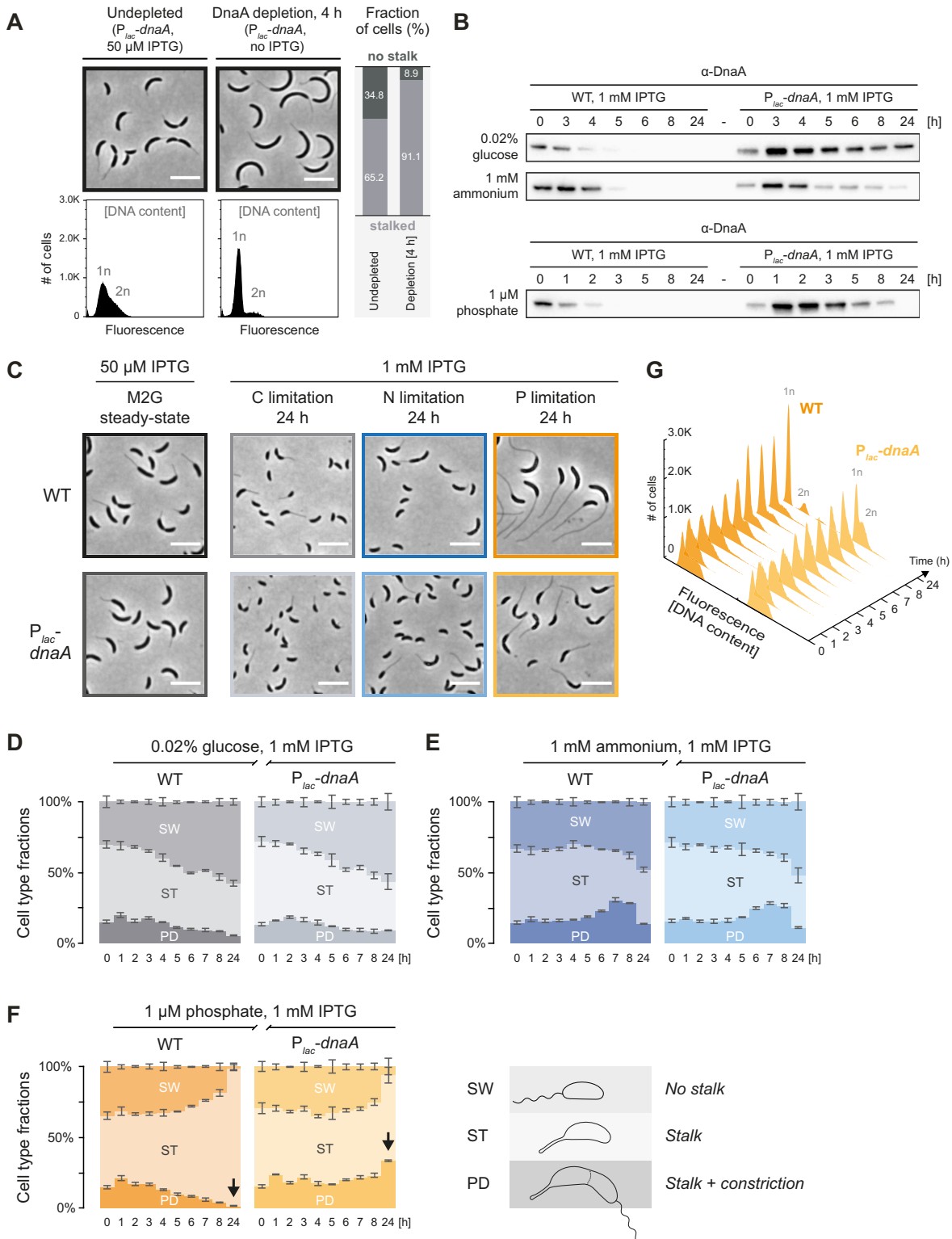

**Fig 5. The P starvation-induced cell cycle arrest can be bypassed by *dnaA* overexpression.** (A) Cell morphology and DNA content of a strain with inducible *dnaA* expession (P$_{lac}$-*dnaA*) grown in M2G under non-depleting conditions (grown with 50 μM IPTG) and under depleting conditions (4 h without IPTG). Quantification of stalked cell fractions was done by manually assigning 1000 cells as either stalked or non-stalked cells, for each condition. The histograms show DNA content of 30 000 cells as measured by flow cytometry. Scale bars: 5 μm. (B) Immunoblots showing DnaA levels of wild-type (WT) and P$_{lac}$-*dnaA* cells in mid-exponential phase in M2G containing

50 μM IPTG (t = 0 h) and after being shifted to nutrient-limited media containing 1 mM IPTG to induce *dnaA* overexpression. **(C)** Micrographs of cells in nutrient-replete cultures with 50 μM IPTG, and after 24 hours in nutrient-limited media containing 1 mM IPTG to induce *dnaA* overexpression. **(D–F)** Stacked bar chart showing the proportions of swarmer cells (SW), stalked cells (ST), and predivisional (PD) cells in wild-type or P$_{lac}$-*dnaA* cultures in mid-exponential phase in M2G containing 50 μM IPTG (t = 0 h) and after shifting cells to nutrient-limited media containing 1 mM IPTG to induce *dnaA* overexpression. Each bar represents the average with error bars showing standard deviations (t = 0 h: six and nine biological replicates for wild-type and P$_{lac}$-*dnaA* respectively; t > 0 h: two and three biological replicates for wild-type and P$_{lac}$-*dnaA* respectively). For each biological replicate, 917–11 020 cells were counted (average: 2911 cells). **(G)** DNA content as determined by flow cytometry of wild-type cells and P$_{lac}$-*dnaA* cells in mid-exponential phase in M2G containing 50 μM IPTG (t = 0 h) and after being shifted to P-limited medium containing 1 mM IPTG to induce *dnaA* overexpression. Each separate histogram represents 30 000 cells. 1n and 2n denote one and two chromosome equivalents respectively.

*crescentus* shows distinct developmental responses under C, N, and P starvation conditions, promoting the accumulation of either motile swarmer cells or sessile stalked cells. Our data reveal that regulation of the coordination between the SW→ST cell differentiation and the G$_1$→S transition, i.e., DNA replication initiation, accounts for these condition-dependent differences (**Fig 6**). Under optimal conditions, as well as under C and N starvation, the SW→ST cell differentiation and the G$_1$→S transition are tightly linked. That is, when a swarmer cell differentiates into a stalked cell, it also initiates DNA replication; *vice versa*, when the SW→ST cell differentiation is blocked, the G$_1$→S transition is inhibited as well. However, the situation is different under P starvation. Here, the G$_1$→S transition is inhibited despite continued SW→ST cell differentiation, thus resulting in G$_1$-arrested stalked cells.

Our work shows that these differences in how the SW→ST cell differentiation and the G$_1$→S transition are coordinated can be explained by two separable control mechanisms. One of them involves the (p)ppGpp-independent regulation of DnaA abundance and DNA replication initiation. This mechanism monitors the availability of major nutrients and downregulates DnaA to help ensure a G$_1$ arrest under a wide range of starvation conditions. Largely independent of this DnaA-centered regulation, a (p)ppGpp-dependent mechanism precisely controls the SW→ST cell differentiation. Although the exact mechanism by which (p)ppGpp blocks the SW→ST cell differentiation remains to be elucidated, previous studies suggest that it involves the signal transduction pathways ultimately regulating CtrA activity and stability [40,42,52]. Under conditions of high (p)ppGpp concentrations, i.e., under C and N starvation, this mechanism hinders the differentiation into stalked cells and provides an additional layer of DNA replication control by maintaining CtrA binding at the origin. The use of two distinct mechanisms that control the G$_1$→S transition and the SW→ST cell differentiation independently, provides a way to temporarily decouple these processes under certain conditions (e.g., P starvation) to favor the accumulation of distinct cell types.

To assign cell types over the course of starvation conditions, we have developed a new machine-learning based workflow that makes use of the image analysis tools ilastik and MicrobeJ. This approach allowed us to quantify the proportions of cell types based on phase-contrast microscopy images, in high throughput, and is thus a valuable tool for future studies. Although we have found our approach to be reliable (**S9 Fig**), we cannot rule out that early stalked cells still lacking a visible stalk are in some cases assigned as swarmer cells.

## Formation of non-replicating stalked cells under P starvation

Our work shows that downregulation of DnaA in combination with unrestricted SW→ST cell differentiation leads to the accumulation of G$_1$-arrested stalked cells under P starvation. Absence of DnaA and DNA replication results in the inability to complete the cell cycle due to the lack of a second copy of the chromosome that is required for cell division septum placement [53] and CtrA reactivation [28]. Since cell growth continues at a low rate under P starvation, these mechanisms lead to elongated stalked cells that lack division septa and show

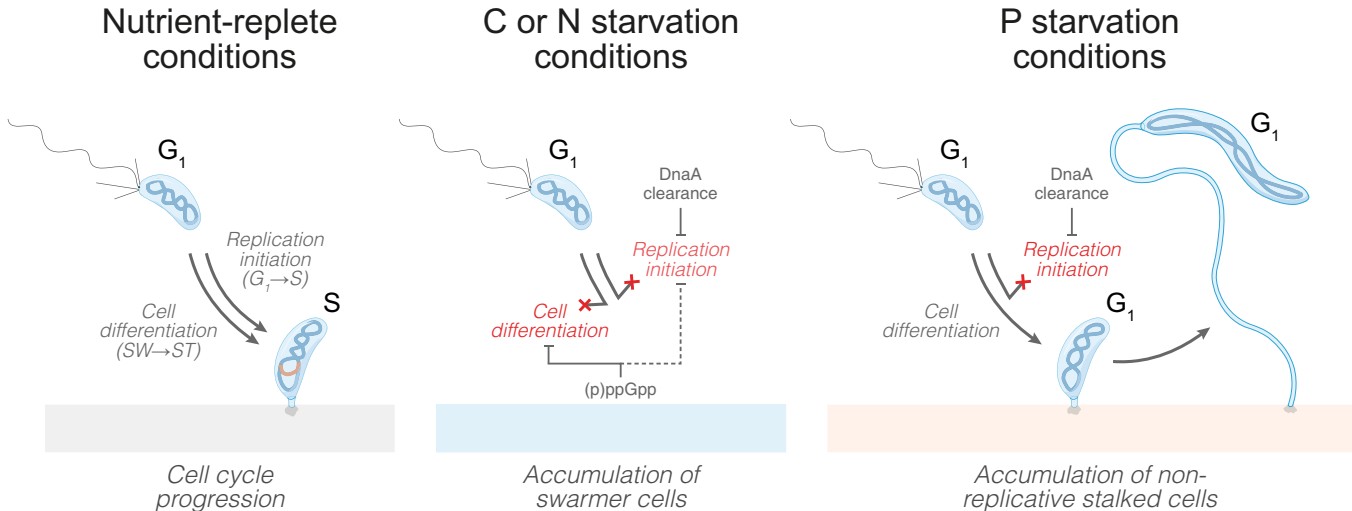

**Fig 6. C, N and P starvation result in the accumulation of distinct cell types due to separable mechanisms acting on cell differentiation and DNA replication initiation.** Under nutrient-replete growth conditions, when a swarmer cell differentiates into a stalked cell it also enters S phase. This coupling of the SW→ST cell differentiation to the $G_1$→S transition is maintained under C or N starvation conditions, during which both processes are blocked through two largely independent mechanisms. Firstly, a (p)ppGpp-dependent mechanism inhibits the SW→ST cell differentiation and may also maintain CtrA-dependent repression of DNA replication (dashed arrow). Secondly, a (p)ppGpp-independent mechanism inhibiting DnaA translation leads to DnaA clearance and a block of DNA replication initiation. In contrast to C and N starvation, under P starvation (p)ppGpp do not accumulate and cells are still able to differentiate into stalked cells. However, DnaA is still cleared through the same mechanism as under C and N starvation, leading to the decoupling of the SW→ST cell differentiation from the $G_1$→S transition and consequently the whole cell population transitioning into $G_1$-arrested stalked cells. The blocked replication initiation prevents cell division, which in combination with continued cellular growth and dramatic stalk elongation, gives rise to characteristic filamentous cells with long stalks.

reduced levels of CtrA. Extended P starvation can lead to drastically elongated cells that display a helical shape when P starvation is combined with increased ammonium levels and alkaline pH, conditions that naturally occur in combination during algal bloom seasons in freshwater habitats [9].

Although our data indicate that the P starvation-induced stalked cell accumulation in *C. crescentus* results from unhindered SW→ST cell differentiation, it is possible that P starvation actively stimulates this differentiation. For example, P starvation may trigger the accumulation of cyclic diguanylate (c-di-GMP), which is known to actively promote the differentiation to stalked cells, partly by driving the dephosphorylation and degradation of CtrA [54–56], thus acting antagonistically to (p)ppGpp [57,58]. A link between P starvation and c-di-GMP signaling has not yet been established in *C. crescentus*. However, previous data indicate that P starvation leads to strong upregulation of the uncharacterized canonical diguanylate cyclase CCNA_03191 [59,60]. We therefore consider this diguanylate cyclase to be an attractive candidate for a starvation-induced regulator of cell development and cellular lifestyle decisions.

## DnaA regulation constitutes a key regulatory step during starvation responses

Although DnaA plays a major role in controlling bacterial cell cycle progression [61], how this protein is regulated in response to nutrient availability remains enigmatic in most bacteria. Our work underscores the regulation of DnaA abundance as a key response during starvation, that not only determines the replication status of cells, but also their developmental state. In *C. crescentus*, inhibition of DnaA synthesis in combination with rapid constitutive DnaA proteolysis ensures fast DnaA clearance, independently of (p)ppGpp signaling [33,34]. Initial work indicated that DnaA, in contrast to C and N limitation, is not cleared under P starvation [31].

However, this study utilized an unconventional P starvation procedure, shifting cells to an unbuffered minimal medium completely lacking P. In the present study, we show that P limitation indeed leads to rapid clearance of DnaA, which is consistent with the work of Billini et al. [35] that reported DnaA clearance under P starvation. In a previous study, we demonstrated that an mRNA sequence downstream of the *dnaA* start codon ($Nt_{DnaA}$) confers translational inhibition under C starvation [34]. Our new work revealed that DnaA regulation occurs by a similar mechanism under N and P starvation. The observation that $Nt_{DnaA}$ is responsive to a wide range of starvation conditions suggests that it monitors the general growth status of cells, rather than the absence or presence of a specific metabolite that is linked to a specific starvation condition. Further work will be necessary to pinpoint how $Nt_{DnaA}$ precisely tunes DnaA synthesis in response to nutrient availability.

### Ecological relevance of sessile and motile behaviors under distinct starvation conditions

The induced motility and increased proportion of swarmer cells observed during C and N limitation may represent an adaptive response helping cells to swim away from nutrient-depleted environments and chemotactically locate new sources of nutrients. Consistently, a study on *C. crescentus* microcolonies showed that removal of the C source stimulates colony dispersal [62]. Moreover, biofilm dispersal in response to C or N starvation is a behavior shared with various bacteria, including pathogens, such as *Vibrio cholerae* [63], *Pseudomonas aeruginosa* [64], *P. fluorescens* [65], and *P. putida* [66]. Remarkably and consistent with our observations of axenic *C. crescentus* cultures, subjecting a mixed population of lake bacteria to C limitation was also shown to produce populations dominated by small motile cells, whereas P limitation led to a strong accumulation of non-motile elongated capsulated cells [67], suggesting that these responses are shared with various environmental bacteria and operate under natural conditions.

If dispersal and motility are common bacterial responses during C and N starvation, why does P starvation instead lead to non-motile stalked cells in *C. crescentus*? In nature, phosphates—the major bioavailable forms of P—are due to their negative charge adsorbed to surfaces of minerals, a process which has large ecological and geochemical consequences [68,69]. Therefore, we speculate that adopting a sessile surface-attaching lifestyle under P starvation could be favorable to bacteria by positioning cells where the limiting nutrient is most abundant. Furthermore, the stalked state itself is likely advantageous for tolerating P starvation, with the stalk probably improving nutrient scavenging [13,14,70]. Stalk elongation in response to P starvation is a conserved feature of the family *Caulobacteraceae* [44,71,72], and also occurs in more distantly related *Alphaproteobacteria* like *Sphingomonas leidyi* (72) and *Rhodomicrobium vannielii* [73]. Additionally, the proportion of stalked cells increase considerably during P starvation in *Brevundimonas subvibrioides* [74], as well as *B. variabilis* and *S. leidyi* (72), species that barely exhibit stalks in complex media. Taken together, the developmental regulation in response to P limitation described in this work may constitute an evolutionarily conserved feature among dimorphic *Alphaproteobacteria*.

## Materials and methods

### Bacterial strains and growth conditions

Bacterial strains, listed in **S1 Table**, were routinely grown on PYE agar [3] and in M2G minimal medium (6.1 mM $Na_2HPO_4$, 3.9 mM $KH_2PO_4$, 9.3 mM $NH_4Cl$, 0.5 mM $MgSO_4$, 0.5 mM $CaCl_2$, 0.01 mM $FeSO_4$ [EDTA-chelated; Sigma, F0518], 0.2% [w/v] glucose) supplemented

with isopropyl β-D-1-thiogalactopyranoside (IPTG; 1 mM for PYE and 50 μM for M2G) or kanamycin (5 μg/ml for liquid and 25 μg/ml for plates) when necessary. Cultures were incubated at 30˚C, in flat-bottomed E-flasks, with 200 rpm orbital shaking. Growth was monitored using V-1200 spectrophotometer (VWR). When appropriate, cultures were diluted with pre-heated fresh medium to maintain mid-exponential growth.

### Nutrient exhaustion assays

Cells were grown in M2G to mid-exponential phase ($OD_{600}$ of 0.2–0.4), before a culture volume corresponding to a final $OD_{600}$ of 0.08 (when diluted to 25 ml) was sampled and centrifuged for 10 min at $7,197 \times g$ in 20˚C. Pellets were washed with nutrient-limited minimal media and centrifugation was repeated. For carbon limitation, M2G with 0.02% (w/v) glucose was used, for nitrogen limitation, M2G with 1 mM $NH_4Cl$ was used, and for phosphorus limitation, M5G [25] with 1 μM or 50 μM total phosphate (6.1:3.9 ratio of $Na_2HPO_4$ and $KH_2PO_4$) was used, as specified. Washed cells were resuspended in 25 ml nutrient-limited medium and put in incubation. Time spent in nutrient-limited medium is counted from the start of incubation.

### Microscopy and image analysis

Cells were imaged directly or fixed with 1% (w/v) formaldehyde and kept at 4˚C until imaging. Fixed cells were imaged within a few hours of sampling to prevent cell sedimentation, as stalks are easily damaged during resuspension of cell sediment. Cells were spotted on 1% (w/v) agarose pads prepared with tap water and phase-contrast images were acquired using an ECLIPSE *Ti* inverted research microscope (Nikon) equipped with a Plan Apo λ 100× Oil Ph3 DM (1.45 NA) objective (Nikon) and a Zyla sCMOS camera (Andor).

To quantify cell type proportions and cell morphology characteristics, the microscopy images were processed by combining a random forest machine learning approach (ilastik, ver. 1.3.3post3) [49] with the FIJI plugin MicrobeJ (ImageJ, ver. 2.1.0; MicrobeJ ver. 5.13l) [50]. ilastik random forest models were trained on 72 phase-contrast microscopy images that were selected to represent most of the variation in the total dataset (differences in image brightness, contrast, sharpness, growth conditions, strains, cell morphology, etc.). Initially, individual pixels of the phase-contrast images were classified (using ilastik pixel classification) according to whether they correspond to a cell body, a stalk or background. Subsequently, the resulting pixel prediction maps were used in the ilastik object classification workflow to segment and classify objects into stalked and swarmer cells. A third object category was used to remove debris and undesirable (i.e., out-of-focus, overlapping, lysed and/or partly obscured) cells from the analysis. Using a separate object classification step, stalks were segmented into objects and classified into two categories to remove undesirable stalks (i.e., out-of-focus, overlapping, and/or partly obscured) from the analysis. In a final step, morphological characteristics (i.e., cell length, stalk length and cell constriction) were obtained using MicrobeJ by segmenting the cells on the phase-contrast images and using the cell type classification and stalk classification from the ilastik object prediction maps to annotate cell types and remove undesirable cells and stalks. For each biological replicate, a total of ten fields of view (133.12 × 133.12 μm) were analyzed per time point. Example images from the workflow can be found in **S9A and S9B Fig**. All files necessary to run the image analysis workflow, including example data as well as training data have been made publicly available at the Swedish National Data Service (SND-ID: 2023–202), accessible at: https://doi.org/10.58141/wfc6-qh32.

Accuracy of the cell type prediction into "Stalked", "Swarmer" and "Debris" by the ilastik random forest model (used for **Figs 1F**, **4A**, and **5D–5F**) was assessed by selecting a subset of

images that capture the entire range of morphological variation seen in this study. Specifically, a total of 24 phase-contrast images of *C. crescentus* WT and Δ*rel* cells in exponential phase growth and after 24 hours in C-, N- and P-limited conditions (pictures included in the **Figs 1F** and **4A** datasets) were selected for validation. Cell type predictions of objects in these images were compared to manual annotations by two *C. crescentus* experts (**S9C–S9E Fig**). Note that a small number of objects were automatically excluded from validation due to lacking expert annotation, leading to small discrepancies in the total number of compared observations shown in **S9C–S9E Fig**. Based on these validation efforts, we can determine that the prediction model has a high accuracy in distinguishing "Swarmer" and "Stalked" classes, but that there is disagreement among the experts and model in determining the confines of the "Debris" class (**S9C–S9E Fig**). Indeed, accuracy scores for the model predictions of only the "Swarmer" and "Stalked" classes are approximately 94.4% and 94.5% when compared to Expert 1 and Expert 2, respectively (**S9C and S9D Fig**) and there is approximately 97.2% expert agreement for these classes (**S9E Fig**). The degree of uncertainty in classifying objects as "Debris" was expected and can be mainly attributed to the subjectivity in determining when cells are sufficiently out of focus to be excluded from further analysis. Importantly, the determination by the model should be consistent over the entire analysis and we could not find noteworthy biases in whether the model keeps or excludes stalked or swarmer cells compared to the expert annotation (**S9C and S9D Fig**).

## Flow cytometry

Cells were fixed in 70% (v/v) ethanol and stored at 4˚C. Samples were prepared for flow cytometric quantification of cellular DNA content as described earlier [33], but using 0.25 μM SYTOX Green (Invitrogen) for staining. Cells were run on a CytoFLEX (Beckman Coulter) flow cytometer (Gains: FSC = 143, SSC = 74, and FITC = 120; Thresholds: SSC-A > 500, FITC-A > 5000; Gating: SSC-A between $10^2$–$10^7$, FITC-A between $2\times10^4$–$10^6$) or an LSR-Fortessa (BD Biosciences) flow cytometer (for **S7B Fig** only; Voltages: FSC = 621, SSC = 302, FITC = 520; Threshold: FITC > 3000) and data was plotted in FlowJo (ver. 10.7.1). Histograms were converted into isometric projections in Adobe Illustrator (Adobe) by a combination of the Scale tool (non-uniform; horizontal: 82%, vertical: 70.5%), the Shear tool (−30˚), and the Rotate tool (−30˚).

## Immunoblot analysis and *in vivo* protein degradation assays

Culture samples of 1000 μl were centrifuged at 12,000 × *g* and pellets were stored at −20˚C. Pellets were resuspended on ice in 200 μl Laemmli sample buffer (2% [w/v] SDS, 10% glycerol, 60 mM Tris-Cl [pH 6.8], 0.01% [w/v] bromophenol blue, 1% β-mercaptoethanol) per $OD_{600}$ unit and denatured at 98˚C for 10 min. Samples were fractionated by SDS-PAGE on 4–20% Mini-PROTEAN TGX Stain-Free protein gels (Bio Rad), and transferred by semi-dry blotting (Trans-Blot Turbo transfer system; Bio Rad), as per manufacturer's instructions, to Amersham Protran 0.45 μm nitrocellulose membranes (GE healthcare). Gels and membranes were imaged using a Gel Doc XR+ imager (Bio Rad) before and after transfer, respectively, to ensure equal sample loading. Blocking was done in TBS buffer with 5% (w/v) skimmed milk powder, before the detection of proteins using primary antibodies against DnaA [18] or CtrA (kindly provided by M. Laub) diluted 1:5000 in TBST (TBS containing 0.1% Tween-20) with 3% skim milk powder, followed by 1:5,000 of goat anti-rabbit secondary horseradish peroxidase-conjugated antibody (Thermo Fisher Scientific #31460) in TBST 0.3% skim milk powder. Blots were developed using SuperSignal Femto West reagent (Thermo Fisher Scientific #34094), and scanned with a LI-COR Odyssey Fc imaging system. Bands were quantified in ImageStudio

(ver. 5.2.5; LI-COR), using box thickness 5 top/bottom segments for average background subtraction.

For *in vivo* protein degradation assays, cells were grown at suitable conditions before protein synthesis was inhibited by addition of 100 μg/ml chloramphenicol dissolved in ethanol. Immunoblot samples were taken right before and every 10 min after chloramphenicol addition until the 60 min mark, by centrifugation at $21,100 \times g$ for 1 min, before snap-freezing pellets in liquid nitrogen.

## Motility assays

Soft agar plates were cast with 0.3% (w/v) Bacto agar (Difco, BD), by mixing pre-heated 2×-concentrated M2G- or M5G-based media with 0.6% (w/v) molten agar at 50˚C. Cells were grown in M2G to mid-exponential phase, before being centrifuged for 5 min at $10,000 \times g$, 20˚C, and resuspended in fresh M2G to a final $OD_{600}$ of 0.1. Plates were injected twice with 1 μl cell suspension, per strain, 4 hours after plate casting, and incubated at 30˚C for 72 hours. Each plate consequently had four spaced-out inoculation sites: two for wild-type and two for Δ*rel*. To control for batch differences, the experiment was done on three separate days, and two separately prepared plates were cast and inoculated for each nutrient composition on each day. Plates were photographed using a Gel Doc XR+ imager (Bio Rad), using white epi illumination without a filter. Motility zone (soft agar colony) areas were quantified in Fiji (ImageJ, ver. 2.1.0) by fitting the largest possible ellipse inside each motility zone using the "Oval" tool. Irregular zones resulting from satellite colony interference were discarded. Each data point in **Figs 1G** and **4B** represents the average motility zone area per day.

## *In vivo dnaA* expression reporter assays

The monitoring of eGFP expressed from *dnaA* expression reporter constructs, including data analysis, was done exactly as described in Felletti et al. [34], but using a detector gain of 80 for P limitation experiments. Since fluorescence measurements are noisy and unreliable for the first 5 hours of an experiment, we opted to using M5G medium with 50 μM phosphate instead of 1 μM to delay the onset of starvation to occur past the period of noisiness (**S10 Fig**).

## Statistical analysis

Statistical analyses (ANOVA, pairwise post hoc tests, Bonferroni corrections, and the appropriate tests to test for underlying assumptions) were carried out using the R open-source software (ver. 4.2.2) [75]. Differences were regarded as significant when the P value was $\leq 0.05$.

## Supporting information

**S1 Fig. Growth and cell morphology after extended incubation in P-limited media. (A)** Growth curves of wild-type cells in nutrient-replete M2G medium (black), and media limited for C (gray), N (blue), or P (orange). **(B)** Micrographs of cells in nutrient-replete M2G medium or after incubation in M5G 1 μM phosphate for the indicated amount of time. Scale bars: 5 μm.
(PDF)

**S2 Fig. Motility assay example pictures. (A)** Representative motility zone (soft agar colony) pictures in grayscale and the *spectrum* color lookup table of ImageJ (ver 2.1.0) after 72 hours incubation. All pictures are in scale (scale bar: 10 mm). **(B)** Micrographs of cells withdrawn from motility agar colony edges after 72 hours incubation, exhibiting clear phosphate

starvation phenotypes in M5G prepared without phosphate. Scale bars: 5 μm.
(PDF)

**S3 Fig. Lon-mediated DnaA proteolysis. (A)** DnaA *in vivo* stability assay of wild-type and *lon*::Ω cells in mid-exponential phase in M2G and while experiencing N or P starvation. **(B)** Quantification of DnaA *in vivo* stability assays done as shown in (A), from either five (M2G mid-exponential phase), two (N-limited), or three (P-limited) independent replicates for the *lon*::Ω strain, shown alongside wild-type data from **Fig 3B**, with error bars showing standard deviations. **(C–E)** Immunoblots of DnaA after shifting cells to nutrient-limited media, comparing *lon*::Ω to wild-type cells. For (C) and (D), the wild-type half of the blot was separated from the *lon*::Ω half, washed, and redeveloped to prevent the high signal of the *lon*::Ω bands from drowning out the wild-type signal. **(F)** DNA content as determined by flow cytometry of wild-type and *lon*::Ω cells after being shifted to C- or P-limited media. Each separate histogram represents 30 000 cells. **(G)** DNA content presented essentially as in (F), of wild-type and *lon*::Ω cells in mid-exponential phase in M2G before being shifted to N-limited medium (t = 0 h), and after being shifted to N-limited medium.
(PDF)

**S4 Fig. The synthesis inhibition conferred by Nt$_{DnaA}$ is independent on the 5′UTR. (A)** Schematic depiction of *dnaA* expression reporter constructs from Felletti et al. [34]. **(B–C)** Growth curves (gray, blue, and orange) and OD$_{600}$-normalized eGFP fluorescence (green) of cells harboring the *dnaA* expression reporter constructs depicted in (A), after shift to C-, N-, or P-limited media. Light hues: Nt$_{dnaA}$-eGFP constructs; dark hues: constructs with eGFP alone. eGFP-coding sequences are preceded by 5′ untranslated regions (5′UTRs) either of the artificial 6/13 UTR type (B), or from the *lac* operon of *E. coli* (C).
(PDF)

**S5 Fig. Growth and morphology of Δ*rel* cells. (A)** Micrographs of Δ*rel* cells sampled from a nutrient-replete culture exponentially growing at steady-state, and cells sampled after 24 hours in nutrient-limited media. Wild-type cells are shown for comparison. Scale bar: 5 μm. **(B)** Immunoblot of DnaA from nutrient-replete wild-type and Δ*rel* M2G cultures in mid-exponential phase (t = 0 h) and after shifting cells to N-limited media. The blot picture has been cropped into two halves to juxtapose the two strains. **(C)** Quantification of DnaA levels in N-starved Δ*rel* cultures (n = 3) compared to wild-type (n = 5). Mean DnaA levels are shown alongside growth curves of the Δ*rel* strain in nutrient-replete minimal medium (n = 5), media limited for N (n = 10), and wild-type growth curves shown in **Fig 1B**. Error bars represent standard deviation. Δ*rel* immunoblot samples were codeveloped with wild-type samples, and band intensities were normalized to t = 0 h bands of the wild type.
(PDF)

**S6 Fig. IPTG titration for the P$_{lac}$-*dnaA* strain. (A)** DnaA immunoblot and respective quantification from wild-type and P$_{lac}$-*dnaA* cells sampled at 4 hours after being shifted to M2G medium containing the indicated IPTG concentrations. Before shift, cells were cultivated until mid-exponential phase in M2G 50 μM IPTG. **(B)** Length measurements of 800 cells sampled as in (A). Lines and error bars indicate means ± SD. The extended horizontal line emphasizes the average length of wild-type cells in the absence of IPTG. **(C)** Micrographs and flow cytometry DNA content histograms (of 30 000 cells each) from cells sampled as in (A). Scale bars: 5 μm.
(PDF)

**S7 Fig. Induction of *dnaA* expression from P*lac* under starvation conditions. (A)** DNA content as determined by flow cytometry of wild-type cells and P*lac*-*dnaA* cells in mid-exponential phase in M2G containing 50 μM IPTG (t = 0 h) and after being shifted to nutrient-limited media containing 1 mM IPTG to induce *dnaA* overexpression. P starvation data is also shown in **Fig 5G**. Each separate histogram represents 30 000 cells. **(B)** DNA content presented as in (A), of a P*lac*-*dnaA* culture being split after 24 hours in P-limited medium containing 50 μM IPTG (basal *dnaA* expression), with one culture half being supplemented with IPTG to a final concentration of 1 mM to boost *dnaA* expression. **(C)** Stacked bar chart showing the proportions of swarmer cells (SW), stalked cells (ST), and predivisional (PD) cells in P*lac*-*dnaA* cultures treated as in (B). Each bar represents the average of two biological replicates with error bars showing standard deviations. For each biological replicate, 586–1141 cells were counted (average: 801 cells). **(D)** Immunoblot showing DnaA levels of P*lac*-*dnaA* cells in mid-exponential phase in M2G containing 50 μM IPTG (0 h in P-limited medium) and after being treated as in (B).
(PDF)

**S8 Fig. *dnaA* overexpression inhibits stalk elongation under P starvation.** Stalk length violin plot of wild-type and P*lac*-*dnaA* cells in mid-exponential phase in M2G or after 24 h in P-limited medium in the presence of IPTG. Concentrations of the P*lac* inducer IPTG are indicated. Shown as done in **Fig 1E**, alongside stalk measurements of wild-type cells grown in absence of IPTG also shown in **Fig 1E**. Stalk measurements of cells grown in M2G containing 50 μM IPTG were done on six and nine biological replicates for wild-type and P*lac*-*dnaA* respectively, and for two and three biological replicates, respectively, for P-limited medium containing 1 mM IPTG.
(PDF)

**S9 Fig. ilastik example images and validation. (A)** Example images showing outputs from each major step of the machine learning-based automatic cell annotation of microscopy pictures. In step 1, pixels of phase-contrast images are classified as 'background', 'cell body', or 'stalk' in ilastik. In step 2, resulting pixel prediction maps are then used in ilastik to predict cell type objects and stalk objects, in two separate procedures. In step 3, cells are segmented and assigned constriction sites in MicrobeJ using the phase-contrast images. Segmented cells are then assigned cell types in MicrobeJ based on the cell type object prediction bitmaps generated by ilastik. Finally, stalks are assigned to the poles of segmented cells in MicrobeJ, using the stalk object prediction bitmaps generated by ilastik. Scale bar: 5 μm. **(B)** Zoomed-in view of the MicrobeJ output shown in (A), highlighting major annotated features. **(C–E)** Heat maps depicting confusion matrices comparing: (C) the manual annotations by Expert 1 to the predictions of the classification model, (D) the manual annotations by Expert 2 to the predictions of the classification model and (E) the manual annotations by Expert 1 to the manual annotations by Expert 2. Panels C, D, and E are plotted with the same color scale and contain a total of 7274, 7283 and 7275 observations, respectively.
(PDF)

**S10 Fig. Noisy fluorescence measurements at the onset of P starvation can be circumvented by using higher initial P concentrations.** Triplicate growth curves and eGFP fluorescence measurements of the *dnaA* expression reporter construct "UTR-Nt" after shift to M5G with various initial phosphate concentrations. $OD_{600}$-normalized eGFP fluorescence values were divided by the lowest observed $OD_{600}$-normalized fluorescence measurement for each "phosphate concentration data set" (resulting in relative $OD_{600}$-normalized fluorescence), to allow

fluorescence patterns to be compared side-by-side.
(PDF)

**S1 Table. Strains used in this study.**
(PDF)

## Acknowledgments

We thank members of the Jonas group for helpful discussions and Jan Olov Persson and Alexander Crompton for statistical consulting.

## Author Contributions

**Conceptualization:** Joel Hallgren, Michele Felletti, Kristina Jonas.

**Data curation:** Joel Hallgren, Julien Mortier.

**Formal analysis:** Joel Hallgren, Kira Koonce, Michele Felletti, Julien Mortier.

**Funding acquisition:** Michele Felletti, Kristina Jonas.

**Investigation:** Joel Hallgren, Kira Koonce, Michele Felletti, Eloisa Turco.

**Methodology:** Joel Hallgren, Kira Koonce, Michele Felletti, Julien Mortier.

**Project administration:** Kristina Jonas.

**Software:** Julien Mortier.

**Supervision:** Joel Hallgren, Michele Felletti, Kristina Jonas.

**Visualization:** Joel Hallgren.

**Writing – original draft:** Joel Hallgren, Kristina Jonas.

**Writing – review & editing:** Joel Hallgren, Kira Koonce, Michele Felletti, Julien Mortier, Kristina Jonas.

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
