## [Decision Letter · Decision Letter 0]

30 Aug 2023

Dear Dr Jonas,

Thank you very much for submitting your Research Article entitled 'Phosphate starvation decouples cell differentiation from DNA replication control in the dimorphic bacterium Caulobacter crescentus' to PLOS Genetics.

The manuscript was fully evaluated at the editorial level and by independent peer reviewers. The reviewers appreciated your careful work and attention to an important topic.  They also identified some concerns that we ask you address in a revised manuscript.

We therefore ask you to modify the manuscript according to the review recommendations. Your revisions should address the specific points made by each reviewer.

Yours sincerely,

Aretha Fiebig, PhD

Guest Editor

PLOS Genetics

Three peer reviewers were enthusiastic about your manuscript and had several suggestions that you could consider to enhance the work. Although it is not necessary to experimentally address all of the suggestions, I recommend prioritizing the suggestions that further support the distinction between arrest of cell cycle and differentiation by manipulating DnaA and/or ppGpp levels in different starvation regimes. For example, two reviewers suggested testing whether concomitant P limitation and ppGpp induction would inhibit the developmental SW-ST transition. Similarly, reviewers were curious about whether DnaA induction could rescue DNA replication in C/N limited cells, or after long term P starvation. Finally, reviewer 2 raised a concern about the interpretation of developmental state of nutrient starved cells by light microscopy alone. This could be addressed by EM, by methods to stain flagella (eg. PMID: 22447900 or PMID: 30079982), or at a minimum by discussing the limitations of the methodology. I look forward to receiving your revised manuscript.

Reviewer's Responses to Questions

**Comments to the Authors:**

Reviewer #1: In this report by Hallgren, et al., the authors investigate the distinct responses to carbon, nitrogen, and phosphate limitation in Caulobacter crescentus. Cell cycle progression in Caulobacter involves both the regulation of DNA replication and the differentiation from stalked to swarmer cells. Here the authors demonstrate that limitation of any of these key nutrients results in translation-inhibition of the DNA replication regulator DnaA. Carbon and nitrogen starvation additionally lead to the production of the alarmone (p)ppGpp which inhibits the swarmer-to-stalk transition and leads to an accumulation of swarmer cells; by contrast, phosphate starvation does not increase (p)ppGpp and the cells overwhelmingly arrest in the stalked phenotype.

This work convincingly demonstrated that while C, N, and P starvation block DNA replication, cellular differentiation is regulated differently in C/N and P limited cells by (p)ppGpp. The increased motility of C/N starved Caulobacter cells is consistent with the phenotype observed in Vibrio and Pseudomonas species.

Overall, I found the experiments to be well designed and the conclusions followed the data. I do not have any serious concerns but offer the following suggestions that may enhance the manuscript.

1. In Figure 5B, it appears that IPTG-induced DnaA expression is not maintained over the time course for N starved cells. Was this observed consistently? Do you think this is due to overall effects on protein expression under N starvation, or is the induction weakening over time? I imagine the lac-promoter was chosen over xylose or vanillate since they would confound the C-starvation data. Could one of these promoters be used specifically for the N-limitation studies?

2. In thinking about how the C/N and P limitation phenotypes are unique, I am curious if these phenotypes could be interconverted. For example, what would happen if (p)ppGpp levels were increased in P limited cells by overexpressing the constitutively active RelA allele from E. coli?

3. For C/N limited cells, rel deletion increased the proportion of stalked cells. What would be expected if this mutation was combined with DnaA overexpression? Would the cells continue to divide until resources were completely exhausted? Would this lead to cell death rather that growth inhibition?

4. Although these are distinct growth regimes, stationary phase entry shares features with nutrient limitation. The SpdS/SpdR two-component system is reported to be involved in stationary phase entry. Interestingly, these genes are upregulated upon carbon limitation (Britos et al. PLoS One, 2011). Since the signal required for activating SpdS is unknown, it may be interesting to assess whether this pathway is differentially regulated in the various nutrient limitations.

A few minor points:

1. As far as I am aware, in Caulobacter papers the enzyme required for (p)ppGpp production is referred to as SpoT. For consistency with existing literature, I would recommend replacing “Rel” with SpoT.

2. Line 277: The figure reference should be S5A.

Reviewer #2: In this manuscript, Hallgren et al. characterise the physiological response of Caulobacter crescentus to carbon, nitrogen or phosphorus starvation. As expected, they showed that upon carbon (C) or nitrogen (N) limitation, both cell cycle progression and cellular differentiation are coordinately inhibited with swarmer cells remaining stuck in G1 and not differentiating into replication-competent stalked cells. In contrast, they found that phosphorus (P) depletion led to a cell cycle block similar to the one observed for C or N starvation but did not inhibit cell differentiation. Hence, C. crescentus starved for P accumulated stalked cells that are unable to initiate DNA replication. By determining the cellular levels of DnaA during nutrient exhaustion (C, N or P) and by comparing the WT to the ∆rel strain (unable to synthesise the second messenger (p)ppGpp), the authors concluded that cell cycle is likely controlled by a common mechanism regulating DnaA levels to nutrient availability while cell differentiation is rather under (p)ppGpp control.

Although many data are already separately available or strongly suggested in the literature, such as for example [(p)ppGpp accumulation during C or N but not P starvation, G1 block upon all the starvation conditions, cell differentiation and growth inhibition upon P limitation leading to elongated stalked cells, DNA levels dropping after nutrient depletion], this study has the advantage to bring all the data together facilitating the comparison between the stressful conditions. On the other hand, the difference observed between P and C or N starvation is very interesting, in particular knowing that Caulobacter likely often faces limitation for these macronutrients.

However, I would suggest to play a little bit more with (p)ppGpp and DnaA levels upon nutrient starvation to further support the two mechanisms behind the cell cycle/cell differentiation coordination.

1. Why did the authors evaluate the proportion of chromosome content by flow cytometry in the Plac-dnaA strain only starved for P and not for C or N? It might be interesting to check whether de novo synthesis of dnaA can bypass the cell cycle block observed in C- or N-deplete conditions as it is the case for P starvation, even if the cell differentiation might still be blocked in a (p)ppGpp-dependent way.

2. Likewise, why didn’t the authors test the ∆lon strain starved for N or P while they showed in a previous report this mutant could bypass the G1 block observed after carbon exhaustion. Measuring the chromosome content by flow cytometry in a ∆lon background after N or P exhaustion might be interesting as well.

3. It might also be interesting to test whether the inducible production of (p)ppGpp could inhibit the swarmer-to-stalked cell transition in P starved cells. This could be achieved either by using the tool developed by the Collier lab (pXTCYC-4-relA’-FLAG) or by concomitantly starving C. crescentus for P and N or C.

4. Gorbatyuk et al (2005) showed that only N or C starvation led to a decrease of DnaA levels, not P limitation. This discrepancy should be at least discussed.

5. Britos et al (2011) showed that carbon starvation uncouples swarmer cell differentiation and the G1-to-S phase transition. They mentioned that upon carbon, the flagellum was ejected and the stalk biogenesis initiated but not completed making the stalk invisible by light microscopy. Hence, swarmer cells starved for carbon were erroneously considered as swarmer cells while in fact these were differentiated stalked cells with the stalks visible by electron microscopy. Since the cell type were determined based on light microscopy images and that cells were considered as swarmer cells because they were stalkless, the authors should check the presence of the flagellum by electron microscopy. And if flagella are detected, the discrepancy between both studies should be discussed.

Minor comments

- p4 (L69): the authors wrote “However, they are arrested in the G1 phase and must differentiate into a stalked cell in order to enter S phase and to complete the cell cycle”. In fact, cell differentiation is not a prerequisite for cell cycle progression, so the swarmer cells do not have to differentiate into stalked cells to enter S phase. There are mutants unable to eject flagellum and/or to synthesise a stalk which can still start DNA replication initiation and ultimately divide.

Please rephrase to remove the mandatory sense of the sentence.

- p12 (L234): the word “weakened” should be replaced by “decreased” or “diminished”.

Reviewer #3: Overall, the work is well executed and clearly presented. It is bizarre that DnaA is cleared so much faster under P limitation compared to C and N limitation without change in stability. The authors showed that for a GFP translational reporter, the 5’UTR is required for downregulation under P starvation, but they have not shown that this region is sufficient and would still confer control under P limitation when grafted downstream of a promoter, other than PdnaA, for example Plac. In fact, Plac expression overrides the P starvation-induced reduction of DnaA to a certain extent, therefore it is evident that promoter control via PdnaA also contributes to DnaA levels and this could be specifically addressed. So far, it is only demonstrated that the UTR confers P-starvation control in the context of the native PdnaA promoter.

I would also like to know if DNA replication can be rescued in P-starved bearing Plac-dnaA, in which IPTG induction occurs after 24h of starvation when the culture only contains 1N cells.

Line 364 “Our new work revealed that DnaA regulation occurs by the same mechanism under N and P starvation.” - And yet the abundance of DnaA during the time course seems substantially different …. ??

**Have all data underlying the figures and results presented in the manuscript been provided?**

Reviewer #1: Yes

Reviewer #2: Yes

Reviewer #3: Yes

PLOS authors have the option to publish the peer review history of their article (what does this mean?). If published, this will include your full peer review and any attached files.

Reviewer #1: No

Reviewer #2: No

Reviewer #3: No

---

## [Decision Letter · Decision Letter 1]

8 Nov 2023

Dear Dr Jonas,

We are pleased to inform you that your manuscript entitled "Phosphate starvation decouples cell differentiation from DNA replication control in the dimorphic bacterium Caulobacter crescentus" has been editorially accepted for publication in PLOS Genetics. Congratulations!

Yours sincerely,

Aretha Fiebig, PhD

Guest Editor

PLOS Genetics

Sean Crosson

Section Editor

PLOS Genetics

Comments from the reviewers (if applicable):

Thank you for your careful consideration of and thorough response to reviewer feedback. All three reviewers were pleased with the revised version. In your final uploaded version please consider the comments of reviewer 3. Additionally, in the final upload, reviewer 1 suggested mentioning in the text the abnormal morphology of the cells in Revision Figure 3 that precluded quantification of cell types.

Reviewer's Responses to Questions

**Comments to the Authors:**

Reviewer #1: The revisions provided by Hallgren et al addressed the majority of my comments and concerns. I appreciate the attempt to express activated RelA (Revision Figure 2). I agree that interpretation of these results (particularly since RelA expression is weak and transient) is difficult and not likely to add to the paper. However, the data for the delta-relA/dnaA overexpression (Revision Figure 3) might be more amenable to further analysis. While the filamentation phenotype may be difficult to quantify using ilastik, the DnaA overexpression alone was analyzed in Figure 5 despite having these long cells. It was not clear to me why similar measurements could not be made in the relA background. Additionally, even though it would be somewhat tedious, these quantifications could be made manually.

Overall, the new data and the responses to the other reviewers did significantly improve the manuscript and I found the mechanistic insights into the differences in cellular responses to C, N, or P starvation to be quite compelling and of interest to the broader microbiology community.

Reviewer #2: All the points highlighted by the reviewers have been properly addressed and new data requested by the reviewers have been incorporated in the revised manuscript, further supporting the main conclusions. In the present form, I do not have any other concern or question.

Reviewer #3: l 253

isn't it formally possible that another protease contributes to DnaA degradation under P starvation, and that this activity could explain the disappearance of Dana in P-starved Lon mutants?

l 276/277 I disagree with this statement, in this work the authors have shown that the 26 codons encoding the DnaA N-terminus (NtDnaA) are required for translation control, but not that they are sufficient for it which is implied by their sentence. Please modify.

I have no objections otherwise.

**Have all data underlying the figures and results presented in the manuscript been provided?**

Reviewer #1: Yes

Reviewer #2: Yes

Reviewer #3: None

PLOS authors have the option to publish the peer review history of their article (what does this mean?). If published, this will include your full peer review and any attached files.

Reviewer #1: No

Reviewer #2: No

Reviewer #3: No

**Data Deposition**

http://datadryad.org/submit?journalID=pgenetics&manu=PGENETICS-D-23-00834R1

**Press Queries**

---

## [Editor Report · Acceptance letter]

22 Nov 2023

PGENETICS-D-23-00834R1 

Phosphate starvation decouples cell differentiation from DNA replication control in the dimorphic bacterium Caulobacter crescentus 

Dear Dr Jonas, 

We are pleased to inform you that your manuscript entitled "Phosphate starvation decouples cell differentiation from DNA replication control in the dimorphic bacterium Caulobacter crescentus" has been formally accepted for publication in PLOS Genetics! Your manuscript is now with our production department and you will be notified of the publication date in due course.

With kind regards,

Anita Estes

PLOS Genetics

On behalf of:
